



# Topographic stresses affect stress changes caused by megathrust earthquakes and condition aftershock seismicity in forearcs: Insights from mechanical models and the Tohoku-Oki and Maule earthquakes

Armin Dielforder[1], Gian Maria Bocchini[2], Andrea Hampel[1]

[1]Institut für Erdsystemwissenschaften, Abteilung Geologie, Leibniz Universität Hannover, Hannover, 30167, Germany
[2]Institut für Geologie, Mineralogie und Geophysik, Ruhr-Universität Bochum, Bochum, 44801, Germany

*Correspondence to*: Armin Dielforder (dielforder@geowi.uni-hannover.de)

**Abstract.** Aftershocks of megathrust earthquakes at subduction zones may be driven by stresses arising from the topography

of the forearc. However, the effect of topographic stresses on aftershock triggering is quantitatively not well understood and has been neglected in Coulomb failure stress models that assess whether the stress change caused by an earthquake promotes or inhibits failure on nearby faults. Here we use analytical and numerical models to examine the importance of topographic stresses on stress changes caused by megathrust earthquakes in the forearc. We show that the superposition of topographic and tectonic stresses leads to a dependence of the stress change on the stress state of the forearc. The dependence on the

forearc stress state largely determines the coseismic stress change induced by a megathrust earthquake and must be considered when calculating Coulomb failure stress changes. We further show that increases in Coulomb failure stress promoting widespread failure in the forearc are only possible if topographic stresses dominate the regional stress field after the megathrust earthquake. Applying our modelling approach to the 2011 $M_w$ 9.0 Tohoku-Oki and 2010 $M_w$ 8.8 Maule megathrust earthquakes shows that the effect of topographic stresses caused Coulomb failure stress changes of up to ~40

MPa, which promoted the majority of aftershocks in the Japanese and Chilean forearcs. The model results further reveal that the spatial distribution of aftershocks was influenced by local differences in pre-earthquake stress states, fault strength and megathrust stress drop. Our analysis highlights the significance of topographic stresses in Coulomb failure stress calculations, enabling a better estimation of seismic hazard at subduction zones.

## 1 Introduction

The concept of Coulomb failure stress change finds broad application to investigate earthquake-induced stress changes and the triggering of aftershocks in various tectonic settings including continental interiors and active margins (e.g., Bagge et al., 2018; Farías et al., 2011; King et al., 1994; Lin and Stein, 2004; Oppenheimer et al., 1988; Pace et al., 2014; Ryder et al., 2012; Saltogianni et al., 2021; Stein, 1999; Terakawa et al., 2013; Toda et al., 2011a). The Coulomb failure stress change



(ΔCFS) describes the relative change in shear and normal stresses imparted by an earthquake on nearby faults and indicates
whether it promotes (positive ΔCFS) or inhibits (negative ΔCFS) failure (e.g., King et al., 1994; Harris, 1998).

Over the past decades, the development of Coulomb failure stress models addressed various factors, including the effects of
the tectonic regime and regional stress field on Coulomb failure stress changes, as well as the mechanisms influencing stress
changes in the postseismic and interseismic periods, such as viscoelastic relaxation, poroelasticity and pore pressure changes
(e.g., Bagge and Hampel., 2016; 2017; Cocco and Rice, 2002; Freed and Lin, 1998; Hainzl, 2004; Hardebeck, 2014;
Hardebeck et al., 1998; Peikert et al., 2024; Peña et al., 2022; Segou and Parsons, 2020). One aspect that has found no
consideration in Coulomb failure stress models is the dependence of the regional stress field on topographic and tectonic
stresses. It is the purpose of the present paper to show how the superposition of topographic and tectonic stresses at active
margins influences the Coulomb failure stress change caused by megathrust earthquakes in the forearc.

Topographic stresses result from the gradient in potential energy that arises in the gravitational field of the Earth between
areas of lower and higher elevation (e.g., Molnar and Lyon-Caen, 1988). Topographic stresses are particularly relevant at
active continental margins, where the gradient in potential energy imposed by the continental margin relief, i.e. the
difference in elevation between the oceanic trench and the mountains and volcanoes in the upper plate, induces margin-
normal tension in the forearc (e.g., Lamb, 2006; Wang and He, 1999). The margin-normal tension is counteracted by the
shear stress on the megathrust, which causes margin-normal compression (Fig. 1a). To a first approximation, the
superposition of margin-normal tension and margin-normal compression determines the stress field in the forearc. During
subduction earthquakes, the shear stress on the megathrust decreases, which reduces the compression of the forearc and
alters the superposition of stresses (e.g., Dielforder et al., 2023; 2020; Herman and Govers, 2020; Wang et al., 2019; Wang
and Hu, 2006).

The decrease in megathrust shear stress (stress drop) and resulting stress changes may be a main trigger of aftershock
seismicity in the forearc, as indicated by upper-plate normal faulting sequences after large megathrust earthquakes (e.g.,
Asano et al., 2011; Dewey et al., 2007; Farías et al., 2011; Hardebeck et al., 2012, Hasegawa et al., 2012; Ryder et al., 2012;
Yoshida et al., 2012). Normal faulting after the 2011 $M_w$ 9.0 Tohoku-Oki earthquake, Japan, occurred in forearc areas that
failed by thrust faulting before the earthquake (e.g., Hasegawa et al., 2012; Yoshida et al., 2012). The change in fault
kinematics indicates that the Tohoku-Oki earthquake locally reversed the stress state in the forearc, which has been
mechanically explained by the stress changes resulting from the stress drop on the megathrust (e.g., Cubas et al., 2013;
Dielforder et al., 2023; Wang et al., 2019).

The details of stress changes caused by megathrust earthquakes and their potential to trigger aftershocks are, however, still
not fully understood. In particular, assessing earthquake stress changes at subduction zones requires to account for the
superposition of topographic and tectonic stresses and hence to include gravity, forearc topography and megathrust shear
stresses in models. The parameters are not included in common Coulomb failure stress models based on dislocation solutions
for a fault embedded in an elastic half-space (e.g., Lin and Stein, 2004). We therefore use in this study mechanical models
that allow calculating total stresses in consideration of gravity, forearc topography, and megathrust shear stress. We first use





analytical stress solutions of the dynamic Coulomb wedge theory (Wang and Hu, 2006) to describe the main effects of topographic and tectonic stresses on Coulomb failure stress changes in a uniform subduction zone prism representing the
frontal part of a forearc (Fig. 1b). We then use plane-strain finite-element models (Fig. 1c) to investigate Coulomb failure stress changes caused by the 2011 $M_w$ 9.0 Tohoku-Oki and the 2010 $M_w$ 8.8 Maule earthquakes in the Japanese and Chilean forearcs, respectively. The finite-element models allow to investigate the stress changes in consideration of a more complex upper-plate structure and earthquake stress change.

Our analysis uses a two-dimensional simplification of a three-dimensional system, i.e., we investigate stresses in the vertical
plane of cross section normal to the plate margin and do not address aspects of oblique plate convergence. We further assume in the finite-element models a linear-elastic rheology for the forearc and restrict our analysis to the immediate coseismic stress change induced by the earthquake. In the following, we use the rock mechanics convention of defining compressive stresses as positive.

## 2 Coulomb failure stress changes in an idealized Coulomb wedge

### 2.1 Stress in a stable Coulomb wedge

The dynamic Coulomb wedge theory describes the first order mechanics of subduction zone prisms in megathrust earthquake cycles by considering temporal variations in megathrust shear stress (Wang and Hu, 2006). The theory builds on the classical critical taper model (Dahlen, 1984; Zhao et al., 1986) and approximates the outermost part of the forearc as a uniform wedge of density $\rho$ overlying the megathrust (Fig. 1b). The wedge geometry is defined by the surface slope $\alpha$ and basal dip angle $\beta$.
The wedge has an elastic-perfectly Coulomb plastic rheology, i.e., it can be in a stable elastic state and in critical state at Coulomb failure. The Coulomb-plastic rheology is defined by the coefficient of friction $\mu$ and pore fluid pressure ratio

$$\lambda = (P - \rho_w g D)/(\sigma_z - \rho_w g D) \tag{1}$$

where $P$ is pore fluid pressure within the wedge, $\rho_w$ is water density, $D$ is water depth, $\sigma_z$ is stress in z-direction (see Fig. 1b for local coordinates) and $g$ is gravitational acceleration (Dahlen, 1984, Wang and Hu, 2006). The megathrust shear stress
obeys the friction law for a cohesionless fault

$$\tau_b = \mu'_b \sigma_n \tag{2}$$

where $\sigma_n$ is normal stress. Parameter $\mu'_b$ is the effective coefficient of megathrust friction and depends on both the intrinsic friction coefficient $\mu_b$ and the effect of pore fluid pressure ratio $\lambda_b$ in the fault zone, i.e. $\mu'_b = \mu_b(1 - \lambda_b)$. For this study, it suffices to consider only values of $\mu'_b$ without defining $\mu_b$ and $\lambda_b$ separately.






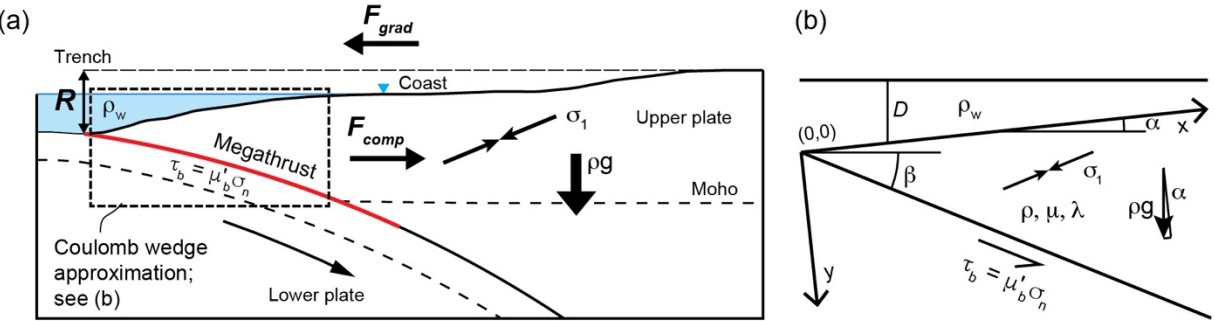

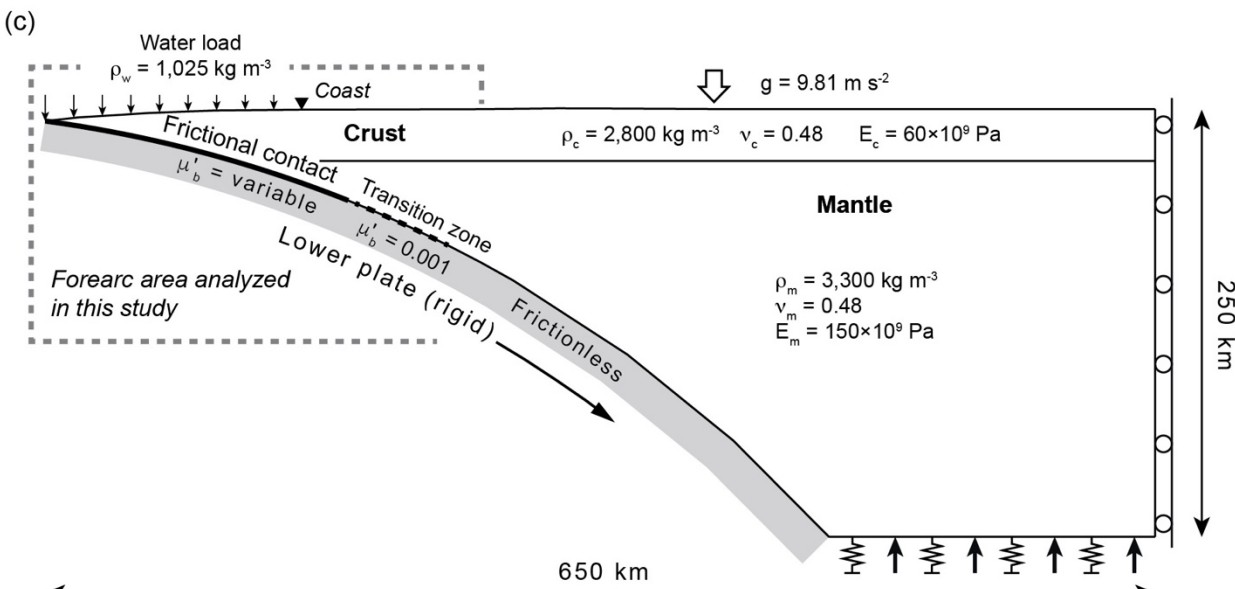

**Figure 1: Conceptual and mechanical models discussed in this study. (a) Main forces determining the stress state in a forearc. Here $\rho$ is rock density, $\rho_w$ is seawater density, *g* is gravitational acceleration, $\sigma_1$ is the greatest compressive stress, $\mu'_b$ is the effective coefficient of megathrust friction, and $\tau_b$ and $\sigma_n$ are the shear and normal stresses on the megathrust. The continental-margin relief, R, imposes a gradient in potential energy that is proportional to the density contrasts between the forearc, seawater and air, and stretches the forearc seawards (force $F_{grad}$). The force $F_{grad}$ is counteracted by the shear stress on the megathrust, which compresses the upper plate (force $F_{comp}$). (b) The Coulomb wedge model discussed in section 2 showing the local coordinate system (x, z). Here *D* is water depth, $\mu$ is coefficient of friction of the wedge material, and $\lambda$ is pore fluid pressure ratio within the wedge, defined by equation (1). (c) Setup and boundary conditions of the finite-element model discussed in section 3. Here $\nu$ is Poisson's ratio and E is Young's modulus. Indices c, m and w indicate crust, mantle and water, respectively.**

The dynamic Coulomb wedge theory reproduces the exact critical stress solutions for noncohesive (Dahlen, 1984) and cohesive (Zhao et al., 1986) Coulomb wedges and additionally provides expressions for stresses in stable Coulomb wedges, as summarised in Appendix 1. For the purpose of describing the dependence of Coulomb failure stress changes on the superposition of tectonic and topographic stresses, it suffices to consider the solutions for a stable noncohesive wedge.




Fig. 2a illustrates the stress in a Coulomb wedge as function of $\mu'_b$ in terms of stresses $\sigma_x$ and $\sigma_z$. The stress solutions are obtained for a reference wedge model with $\alpha = 3°$, $\beta = 10°$, $\mu = 0.7$, and $\lambda = 0$. The dependence of stress on $\mu'_b$ is everywhere the same in the uniform wedge (Dahlen, 1984, Wang and Hu, 2006) and is described in terms of normalized stress values. To allow an estimation of absolute stresses, we provide absolute stress values at a point P1, located at 75 km

from the wedge tip and at 10 km depth on the secondary ordinate (right vertical axis) in Fig. 2a.

**Figure 2: Analytical stress solutions for a stable dynamic Coulomb wedge. (a, b) Solutions for the reference wedge model discussed in the text. (a) Superposition of stresses $\sigma_x$ and $\sigma_z$ as function of effective coefficient of megathrust friction $\mu'_b$. Parameter $\mu'_{b-N}$**
**denotes the $\mu'_b$ value, for which the stress state is neutral ($\sigma_x = \sigma_z$). Left ordinate shows stress values normalized to the maximum value of $\sigma_x$. Right ordinate shows total stresses at point P1 located at 75 km distance from the wedge tip and at 10 km depth. (b) Differential stress ($\sigma_1 - \sigma_3$) and plunge of $\sigma_1$ as function of $\mu'_b$. The differential stress is normalized to the maximum value of ($\sigma_1 - \sigma_3$). (c, d) Normalized ($\sigma_1 - \sigma_3$) and plunge of $\sigma_1$ for different surface slopes $\alpha$ and a basal dip angle $\beta$ of 10°. (e) Normalized ($\sigma_1 - \sigma_3$) and plunge of $\sigma_1$ for different values of $\beta$ and $\alpha = 3°$.**




For low values of μ'ᵦ, σₓ is smaller than σ_z and the wedge is under deviatoric tension (Fig. 2a). The stress state results from the margin-normal tension induced by the topographic relief, which reduces σₓ relative to σ_z. The magnitude of σ_z results from the weight of the overburden. Increasing μ'ᵦ increases the margin-normal compression and decreases the difference between σₓ and σ_z until both stresses are equal. At that point, the stress state is neutral and the compression caused by the megathrust shear stress equals the tension induced by the topographic relief. The value of μ'ᵦ at the neutral stress state is denoted μ'ᵦ-N (Wang and Hu, 2006). If μ'ᵦ is larger than μ'ᵦ-N, σₓ is larger than σ_z and the wedge is under deviatoric compression (Fig. 2a).

Fig. 2b illustrates the same stress dependence in terms of differential stress ($\sigma_1 - \sigma_3$), where $\sigma_1$ and $\sigma_3$ are the greatest and least compressive stresses, respectively. The differential stress is a convex function of μ'ᵦ and is minimal if the stress state is neutral, i.e. if μ'ᵦ = μ'ᵦ-N. The second ordinate in Fig. 2b shows the plunge of stress axis $\sigma_1$ from horizontal. The plunge decreases with increasing μ'ᵦ from ~78° to ~6° and is 45° at the neutral stress state.

The dependence of the stress on μ'ᵦ is similar for every wedge geometry but the value of μ'ᵦ at which the stress state switches from deviatoric tension to deviatoric compression differs with the surface slope and is lower for smaller values of α (Fig. 2c, d). In the absence of surface slope (α = 0), the wedge experiences no deviatoric tension and is always under deviatoric compression for μ'ᵦ > 0. The stress within the wedge also depends on the basal dip angle, but to a lesser extent than on the surface slope (Fig. 2e).

## 2.2 Stress changes caused by megathrust earthquakes

During megathrust earthquakes, the shear stress on the plate interface decreases abruptly due to dynamic weakening processes (e.g., Kanamori and Brodsky, 2004; Scholz 1998; Wang and Hu, 2006). The average stress drop within the rupture zone has been estimated to <10 MPa, although the shear stress may locally decrease or even increase on the rupture surface by a few tens of MPa (e.g., Allmann and Shearer, 2009; Brown et al., 2015; Lee et al., 2011; Luttrell et al, 2011; Kubota et al., 2022; Wang et al., 2020). The average stress drop on the megathrust can be modelled by the dynamic Coulomb wedge theory as a change in the effective coefficient of megathrust friction Δμ'ᵦ = μ'ᵦ-pre − μ'ᵦ-post, where μ'ᵦ-pre and μ'ᵦ-post are the μ'ᵦ values that describe the megathrust shear stress just before and after the earthquake, respectively (Wang and Hu, 2006).

The average stress drop on the megathrust reduces the compression of the wedge. The corresponding stress change depends on μ'ᵦ-pre value due to the superposition of topographic and tectonic stresses as illustrated in Fig. 3 for the reference wedge model and for Δμ'ᵦ = 0.01 (see also orange arrows in Fig. 2b). The Δμ'ᵦ value of 0.01 corresponds to an average stress drop of ~5 MPa at 10-30 km depth. When μ'ᵦ-pre ≤ μ'ᵦ-N, the stress drop on the megathrust increases the differential stress in the wedge because topographic stresses exceed tectonic stresses and become even more dominant (arrows 1 in Fig. 2b; Fig. 3). When μ'ᵦ-pre >> μ'ᵦ-N, the stress drop decreases the differential stress because tectonic stresses become smaller but still exceed




topographic stresses (arrows 2 in Fig. 2b; Fig. 3). When $\mu'_{\text{b-pre}}$ is only slightly larger than $\mu'_{\text{b-N}}$, the megathrust stress drop may increase or decrease the differential stress within the wedge, while the stress state switches from deviatoric compression to deviatoric tension (arrows 3 in Fig. 2b; Fig. 3).


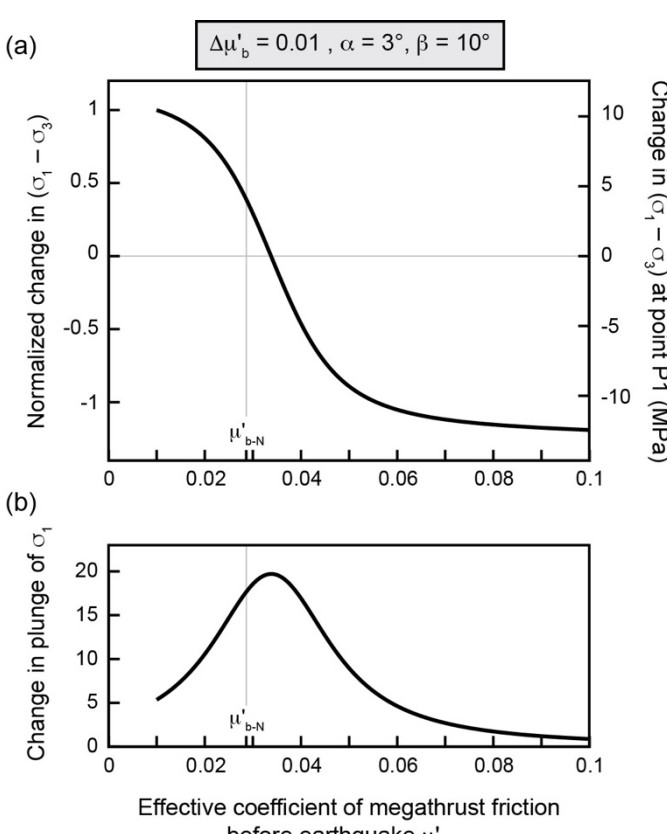

**Figure 3: Stress change due to the stress drop in a megathrust earthquake. The stress drop is given in terms of the change in effective coefficient of megathrust friction $\Delta\mu'_b$, with $\Delta\mu'_b = 0.01$. The stress change depends on the $\mu'_b$ value before the earthquake, $\mu'_{\text{b-pre}}$. The solutions are for the reference wedge model discussed in the text. Parameter $\mu'_{\text{b-N}}$ denotes the $\mu'_b$ value, for**

**which the stress state is neutral. (a) Change in differential stress $(\sigma_1 - \sigma_3)$. Left ordinate shows the change in $(\sigma_1 - \sigma_3)$ normalized to the maximum change in $(\sigma_1 - \sigma_3)$. Right ordinate shows the change in $(\sigma_1 - \sigma_3)$ at point P1 located at 75 km distance from the wedge tip and at 10 km depth. (b) Change in the plunge of $\sigma_1$.**

## 2.2 Assessment of Coulomb failure stress changes in consideration of total stresses

We now examine the stress change due to the stress drop in a megathrust earthquake in terms of Coulomb failure stress change to determine whether it promotes or inhibits failure. The Coulomb failure stress is calculated from the analytical stress solutions as



$$\Delta\text{CFS} = \Delta|\tau| - \mu\Delta\sigma_n = (|\tau_{post}| - |\tau_{pre}|) - \mu(\sigma_{n\text{-}post} - \sigma_{n\text{-}pre}) \tag{3}$$

where $\tau$ is the shear stress and $\sigma_n$ the normal stress on a failure plane (e.g., King et al., 1994; Harris, 1998). The subscript pre

and post denote values before and after the earthquake, respectively. The vertical bars ($||$) indicate absolute shear stress values. The shear and normal stresses are calculated from the analytical stress solutions as

$$\tau = 0.5(\sigma_1 - \sigma_3)\sin 2\omega \tag{4}$$

$$\sigma_n = 0.5(\sigma_1 + \sigma_3) - 0.5(\sigma_1 - \sigma_3)\cos 2\omega \tag{5}$$

where $\omega$ is the angle from the axis of $\sigma_1$ to the failure plane (e.g., Jaeger and Cook, 1979). The shear stress on failure planes

is taken as positive for a sinistral and negative for a dextral sense of shear.

In this study, we determine Coulomb failure stress changes from the differences in shear and normal stresses on failure planes with an optimal orientation to $\sigma_1$ before and after the earthquake, by solving equations (4) and (5) for $\omega = \omega_{opt} = 0.5\tan^{-1}(1/\mu)$ (e.g., Sibson, 1998). The optimal failure planes before and after the earthquake are not identical because the principal stress axes rotate due to earthquake (Figs. 2b, 3b). Our calculations thereby differ from the

conventional approach of resolving the Coulomb failure stress change on a failure plane that is identical before and after the earthquake (e.g., King et al, 1994, Lin and Stein, 2004; Oppenheimer et al., 1988; Stein, 1999). We refer to the Coulomb failure stress changes obtained for $\omega = \omega_{opt}$ as $\Delta\text{CFS}(\omega_{opt})$ (Fig. 4). The $\Delta\text{CFS}(\omega_{opt})$ values indicate whether failure on optimally oriented faults becomes more likely or less likely due to the change in total stress, and can be understood analogously to $\Delta\text{CFS}$ values resolved on a specific failure plane, i.e.:

- If $\Delta\text{CFS}(\omega_{opt}) > 0$, the post-earthquake optimally oriented planes are closer to failure than the pre-earthquake optimally oriented planes, thus failure is promoted (Fig. 4a).

- If $\Delta\text{CFS}(\omega_{opt}) < 0$, the post-earthquake optimally oriented planes are further away from failure than the pre-earthquake optimally oriented planes, thus failure is inhibited (Fig. 4b).

- If $\Delta\text{CFS}(\omega_{opt}) = 0$, the post-earthquake and pre-earthquake optimally oriented planes are equally close to failure,
thus failure propensity remains unchanged (Fig. 4c).

Note that the dashed failure envelopes in Fig. 4 are for visual guidance only and are not intended to imply a critical stress state before the earthquake or similar; the Coulomb failure stress change is independent of the question of criticality. The $\Delta\text{CFS}(\omega_{opt})$ values are identical for failure planes with a sinistral and dextral sense of shear and are not further differentiated hereinafter.






**Figure 4: (a-c)** Mohr's circles illustrating hypothetical stress changes caused by an earthquake and the respective changes in Coulomb failure stress $\Delta CFS(\omega_{opt}) = \Delta|\tau| - \mu\Delta\sigma_n$, where $\mu$ is the coefficient of friction, and $\tau$ and $\sigma_n$ are the shear and normal stresses on a failure plane. Black and orange circles indicate stresses before and after the earthquake, respectively. Angle $\omega_{opt}$ is the angle between the greatest principal stress $\sigma_1$ and optimally oriented failure planes. The failure envelopes (grey dashed lines) are only intended to facilitate the visual assessment of the stress change, but are not meant to imply a critical stress state before the earthquake. **(d)** Sketches illustrating the orientation of failure planes with respect to $\sigma_1$. Angles $\omega_{pre}$ and $\omega_{post}$ indicate the angle between $\sigma_1$ and the failure planes before and after the earthquake respectively. Indices s and d denote the angles for failure planes with a sinistral and dextral sense of shear respectively. For $\Delta CFS(\omega_{opt})$, angle $\omega$ always equals $\omega_{opt}$. The Coulomb failure stress change therefore determines the change in shear and normal stresses between optimal failure planes (grey dash dotted lines) before and after the earthquake, which are not identical because of the rotation of $\sigma_1$. For comparison, the sketch on the right-hand-side shows the conventional calculation of $\Delta CFS$ that determines the change in shear and normal stresses on the optimal failure planes after the earthquake (King et al., 1994). The rotation of $\sigma_1$ causes that $\omega_{pre-d} \neq \omega_{pre-s} \neq \omega_{opt}$.

Fig. 5a illustrates $\Delta CFS(\omega_{opt})$ as function of $\mu'_{b\text{-pre}}$ for the reference wedge model and for $\Delta\mu'_b = 0.01$. The Coulomb failure stress change decreases with increasing $\mu'_{b\text{-pre}}$, with its magnitude and sign being mainly controlled by the stress state in the wedge. The Coulomb failure stress change is positive if the wedge is under deviatoric tension before and after the





earthquake. The state of deviatoric tension promotes normal faulting, which is further promoted by the decrease in horizontal

compression due to the megathrust stress drop. Conversely, the Coulomb failure stress change tends to be negative if the

wedge is under deviatoric compression before and after the earthquake. The state of deviatoric compression allows thrust

faulting in the wedge, but the decrease in horizontal compression inhibits thrust faulting. The Coulomb failure stress change

is further controlled by the change in differential stress. The megathrust stress drop causes an increase in differential stress in

the wedge if $\mu'_{b\text{-pre}} < 0.034$ but a decrease in differential stress is $\mu'_{b\text{-pre}} > 0.034$ (Fig. 3a). Increases and decreases in

differential stress generally tend to promote and inhibit failure, respectively.

For comparison, we also show the Coulomb failure stress change resolved onto the optimally oriented failure planes after the

earthquake following the approach of King et al. (1994), hereafter denoted as $\Delta$CFS(K94) (see right sketch in Fig. 4d). The

$\Delta$CFS(K94) values show a similar dependence on $\mu'_{b\text{-pre}}$ than the $\Delta$CFS($\omega_{opt}$) values but are always slightly higher. For $\mu'_{b\text{-pre}}$

values between 0.045 and 0.049, the $\Delta$CFS values differ in sign, i.e., the $\Delta$CFS(K94) values are positive, while the

$\Delta$CFS($\omega_{opt}$) values are negative. The difference between the values is illustrated in terms of Mohr's circles and for $\mu'_{b\text{-pre}}$ =

0.047 in Fig. 5b. The negative $\Delta$CFS($\omega_{opt}$) value indicates that the total stresses are less favourable for failure after the

earthquake than before the earthquake. Accordingly, the Mohr's circle after the earthquake is farther away from failure than

the Mohr's circle before the earthquake (note that the dashed failure envelope is shown for visual guidance only and is not

intended to imply a critical stress state before the earthquake). By comparison, the positive $\Delta$CFS(K94) value indicates that

the stress change brings the post-earthquake optimal failure planes closure to failure than they were before the earthquake.

Without knowledge of pre-existing weaknesses or strength anisotropies that may favour failure on specific faults, the

$\Delta$CFS($\omega_{opt}$) values better reflect how the change in total stresses alters the proximity to failure. We therefore report hereafter

only $\Delta$CFS($\omega_{opt}$) values. The investigated effect of topographic and tectonic stresses on earthquake induced stress changes is

independent of this choice. Note that for $\mu'_{b\text{-pre}}$ values > 0.08 the $\Delta$CFS($\omega_{opt}$) and $\Delta$CFS(K94) are similar because the plunge

of $\sigma_1$ changes little in the earthquake (Fig. 3b), so that the pre-earthquake and post-earthquake optimal failure planes are

nearly identical.

The Coulomb failure stress change further depends on the surface slope $\alpha$ and to a lesser extent on basal dip angle $\beta$ (Fig. 6a,

b). A steeper surface slope increases the range of $\mu'_{b\text{-pre}}$ values for which the change in Coulomb failure stress is positive. In

the absence of surface slope ($\alpha = 0$), $\Delta$CFS($\omega_{opt}$) is always negative, because the wedge cannot attain a state of deviatoric

tension and normal faulting is never promoted (Fig. 6a). The magnitude of the Coulomb failure stress change scales with the

magnitude of the stress drop, where higher and lower values of $\Delta\mu'_b$ increase and decrease the absolute $\Delta$CFS($\omega_{opt}$) value,

respectively (Fig. 6c). The Coulomb failure stress change increases with the coefficient of friction $\mu$ (Fig. 6d). The

dependence may be surprising because a higher $\mu$ value refers to a stronger fault, which one might expect to be more

difficult to reactivate. However, the stress drop on the megathrust leads to a decrease in normal stress on nearby faults which

tends to promote faulting. The change in normal stress is scaled with parameter $\mu$ when calculating the change in Coulomb

failure stress (term '$\mu\Delta\sigma_n$' in equation (3)), so a higher $\mu$ value yields a larger $\Delta$CFS($\omega_{opt}$) value.



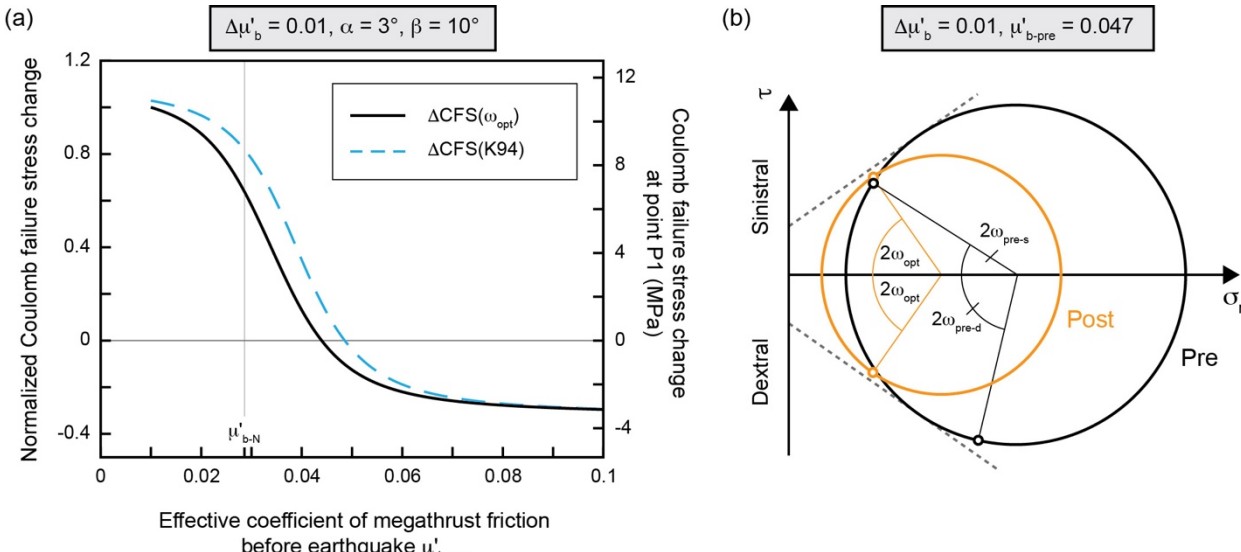

**Figure 5: Coulomb failure stress change due to the stress drop in a megathrust earthquake. Solutions for the reference wedge model discussed in the text. (a) Coulomb failure stress change as function of effective coefficient of megathrust friction before the earthquake, μ'_{b-pre}, and for Δμ'_b = 0.01. Black solid lines indicate ΔCFS(ω_{opt}) values, blue dashed lines indicate conventional ΔCFS(K94) values following King et al. (1994). Left ordinate shows the Coulomb failure stress change normalized to the maximum ΔCFS(ω_{opt}) value. Right ordinate indicates the Coulomb failure stress change at point P1 located at 75 km distance from the wedge tip and at 10 km depth. (b) Mohr's circles, illustrating the stress change for μ'_{b-pre} = 0.047 and Δμ'_b = 0.01. The black and orange Mohr's circles illustrate the stress before and after the earthquake, respectively. The small black and orange circles mark the shear (τ) and normal stresses (σ_n) on the post-earthquake optimal failure planes oriented at angle ω_{opt}. Angles ω_{pre-s} and ω_{pre-d} indicate the angle between post-earthquake optimal failure planes with a sinistral and dextral sense of shear, respectively, with respect to σ_1 before the earthquake (cf. Fig. 4d).**

Figs. 4-6 illustrate Coulomb failure stress changes due to a 'positive' stress drop on the megathrust, i.e., a decrease in megathrust shear stress. A negative stress drop, i.e., an increase in megathrust shear stress, has the opposite effect on the total stresses in the wedge and the corresponding Coulomb failure stress changes (see Fig. S1 in the Supplement). An increase in megathrust shear stress tends to inhibit normal faulting if the wedge is under deviatoric tension and to promote thrust faulting if the wedge is under deviatoric compression. In nature, negative stress drops contribute to arrest the earthquake rupture and may occur downdip and updip of the main rupture area or along velocity strengthening patches enclosed in the main rupture area (e.g., Bilek and Lay, 2002; Brown et al., 2015; Luttrell et al. 2011; Wang and Hu, 2006; Wang et al., 2020).



**Figure 6: Dependence of the ΔCFS(ω_opt) on model parameters. All solutions are for the reference wedge model discussed in the text, but with adjusted values of α, β, μ, and Δμ'_b. (a) Solutions for different values of surface slope α. (b) Solutions for different values of basal dip angle β. (c) Solutions for different stress drops, given in terms of Δμ'_b (change in the effective coefficient of megathrust friction μ'_b). (d) Solutions for different values of coefficient of friction μ. Left ordinate shows ΔCFS(ω_opt) values normalized to the maximum ΔCFS(ω_opt) value obtained for reference wedge model. Right ordinate indicates ΔCFS(ω_opt) values at point P1 located at 75 km distance from the wedge tip and at 10 km depth.**





## 3 Coulomb failure stress changes from force-balance models for forearc stresses

### 3.1 Model setup

We use plane-strain finite-element models of force balance to examine the Coulomb failure stress changes caused by the 2011 $M_w$ 9.0 Tohoku-Oki earthquake, Japan, and the 2010 $M_w$ 8.8 Maule earthquake, Chile. The finite-element models are based on the modelling approaches of Wang et al. (2019) and Dielforder et al. (2023) and yield the total stresses in a forearc resulting from gravity, forearc topography and the shear stress on the megathrust (Fig. 1c). The models are created with the commercial finite-element software ABAQUS (version 2016) and comprise a rigid lower plate in frictional contact with an

elastic upper plate that is subdivided into continental crust and mantle. Following previous studies (Dielforder et al., 2023; Dielforder and Hampel, 2021; Wang et al., 2019; Wang and He, 1999), we adopt a nearly incompressible material for the upper plate (Poisson ratio $\nu$ = 0.48) and densities of 1025 kg/m³, 2800 kg/m³, and 3300 kg/m³ for water, crust, and mantle respectively. The Young's moduli for crust and mantle are 60 GPa and 150 GPa, respectively. We note that using a different Poisson's ratio (e.g., $\nu$ = 0.3) or Young's modulus (e.g., E = 100 GPa for the mantle) makes little difference to the model

results (cf. Dielforder and Hampel, 2021; Wang et al., 2019). A lithostatic pressure and an elastic foundation are applied to the bottom of the model to implement isostasy (arrows and springs in Fig. 1c). The right-hand side of the model (back side of the upper plate) is free to move vertically but is fixed in the horizontal direction. All models are meshed with linear tetrahedral elements with an average element edge length of ~2 km.

We compute individual models for two cross sections across the Japanese and Chilean forearcs, respectively. The cross

sections are oriented perpendicular to the plate margin and cover the area of the megathrust earthquake hypocentre and the area of most intense aftershock seismicity in the forearc (Sendai and Iwaki cross sections in Fig. 7, and Pichilemu and Concepción cross sections in Fig. 8). The models for the different cross sections across the Japanese and Chilean forearcs regard the site-specific margin topography and slab geometry within 50-km-wide swath profiles (dashed rectangles in Fig. 7a, b and 8a, b). The margin topography is approximated by the mean elevation, which we calculate from the ETOPO1

global relief model using TopoToolbox for MATLAB (Amante and Eakins, 2009; Schwanghart and Scherler, 2014). The slab geometry is approximated by fitting an arc with constant curvature through the upper 80 km of the Slab2 model (Hayes et al., 2018).

The megathrust is implemented as a frictional contact between the upper and lower plates and extends from the trench down to a depth of 60 km. The shear stress on the megathrust obeys the friction law for a cohesionless fault (see equation (2)) and

is generated by displacing the lower plate in downdip direction tangential to the plate interface. The displacement ensures that the entire plate interface is at a state of failure (cf. Wang and He, 1999). The stress and strain in the upper plate are independent of the total displacement of the lower plate. The coefficient of megathrust friction can vary along the megathrust, which allows a detailed control on stress magnitudes. In nature, the megathrust transitions into a viscous shear zone downdip of the seismogenic zone, with the shear stress decreasing towards zero with depth (Lamb, 2006; Wada and

Wang, 2009). We implement the transition zone in the models as frictional contact between the downdip limit of the





megathrust and 80 km depth (cf. Dielforder and Hampel, 2021). The $\mu'_b$ value of the transition zone is set to 0.001, which results in a low shear stress of 2-3 MPa. Below a depth of 80 km, the contact between the lower and upper plates is frictionless, i.e., the shear stress on the contact is zero.

Each model run includes three analysis steps. In the first step, a geostatic prestress is assigned, gravity is applied, and
isostatic equilibrium is established. The second and third analysis steps are used to calculate the total stresses in the forearc just before and after the earthquake, similar to the pre-earthquake and post-earthquake steps in the dynamic Coulomb wedge model. In the second analysis step, the $\mu'_{b\text{-pre}}$ values are assigned to the megathrust and the lower plate is displaced. At that stage, ABAQUS yields the total stresses in the upper plate for the pre-earthquake configuration. In the third analysis step, the $\mu'_{b\text{-post}}$ values are assigned to the megathrust, the lower plate is displaced, and ABAQUS yields the total stresses in the upper
plate for the post-earthquake configuration. The Coulomb failure stress change is computed from the model results for step 2 and 3 following equations (3) through (5) and for $\mu = 0.7$.

### 3.2 Pre-earthquake stress state and megathrust stress drop

In section 2, we have shown that the Coulomb failure stress change caused by a megathrust earthquake depends on the pre-
earthquake stress state and hence on the megathrust shear stress before the earthquake, $\tau_{b\text{-pre}}$. We constrain $\tau_{b\text{-pre}}$ using the following procedure (cf. Wang et al., 2019). We first calculate $\tau_{b\text{-pre}}$ using estimates of $\mu'_b$ values derived from heat-dissipation models (Gao and Wang, 2014) and force balance models (Dielforder, 2017; Lamb, 2006). The $\mu'_b$ values estimate the apparent strength of the megathrust, i.e., the level of shear stress that the fault can sustain before great earthquakes (heat-dissipation models) and that is required to prevent the continental-margin relief from gravitational collapse (force-balance
models). Both modelling approaches yield comparable $\mu'_b$ values of about 0.03±0.01 for the Japanese and Chilean megathrusts (Dielforder, 2017; Gao and Wang, 2014; Lamb, 2006). We note that the $\mu'_b$ values are representative for the average of numerous earthquake cycles and are used solely as an initial estimate of $\mu'_{b\text{-pre}}$ to obtain a preliminary value of $\tau_{b\text{-pre}}$.

We then estimate the megathrust shear stress after the megathrust earthquake, $\tau_{b\text{-post}}$, by calculating the stress drop within the
50-km-wide swath profiles from published stress-drop models and subtracting it from the $\tau_{b\text{-pre}}$ values. For the Tohoku-Oki earthquake, we use the stress drop-model of Brown et al. (2015) for the coseismic-slip model of Iinuma et al. (2012) (Fig. 7b, d). For the Maule earthquake, we use the stress-drop model of Luttrell et al. (2011) for the coseismic-slip model of the same authors (Fig. 8b, d). We then solve the finite element model for the $\tau_{b\text{-post}}$ values and evaluate whether the resultant stress state is compatible with focal mechanisms of upper-plate earthquakes that occurred within the 50-km-wide swath
profiles in the first year after the megathrust earthquake (Figs. 7c and 8c). The focal mechanisms are from the Japan Meteorological Agency and from the earthquake catalogue of Şen et al. (2015) for Japan and Chile, respectively.



**Figure 7: Seismotectonic setting of northeast Japan. (a) Seismicity in the upper (North American) plate after the 2011 $M_w$ 9.0 Tohoku-Oki earthquake (yellow star). Grey dots are aftershock hypocentres from the Japan Meteorological Agency (JMA). All events have a magnitude ≥ magnitude of completeness of 1.5 (determined with the maximum curvature method of Wiemer and Katsumata, 1999). Beach balls denote JMA focal mechanism solutions. (b) Earthquake slip contours in meter (pink lines and numbers) and coseismic megathrust stress drop (blue to red colour bar signal). Slip contours and stress drop from Iinuma et al. (2012) and Brown et al. (2015), respectively. (a, b). Black lines indicate location of cross sections shown in (c, d). Dashed rectangles indicate the width of swaths (50 km) used to evaluate the seismicity distribution and fault kinematics (c) and the average stress drop on the megathrust (d) along the Sendai (Se) and Iwaki (Iw) cross sections.**



**Figure 8: Seismotectonic setting of south-central Chile. (a) Seismicity in the upper (South American) plate after the 2010 $M_w$ 8.8**
**Maule earthquake (yellow star). Gray dots are aftershock hypocentres from Lange et al. (2012). All events have a magnitude ≥**
**magnitude of completeness of 2.0 (determined with the maximum curvature method of Wiemer and Katsumata, 1999). Beach balls**
**denote focal mechanism solutions from Şen et al (2015). (b) Earthquake slip contours in meter (pink lines and numbers) and**
**coseismic megathrust stress drop (blue to red colour bar signal). Slip contours and stress drop from Luttrell et al. (2011). (a, b).**
**Black lines indicate location of cross sections shown in (c, d). Dashed rectangles indicate the width of swaths (50 km) used to**
**evaluate the seismicity distribution and fault kinematics (c) and the average stress drop on the megathrust (d) along the Pichilemu**
**(Pi) and Concepción (Co) cross sections.**



We assume that the modelled stress state is compatible with focal mechanism solutions, if thrust faulting and normal faulting events occur in areas of deviatoric compression and deviatoric tension, respectively. If the stress state and focal mechanism are not compatible, we adjust the $\tau_{b\text{-pre}}$ values by changing parameter $\mu'_{b\text{-pre}}$ and repeat the subsequent calculations. We repeat the procedure until the stress state after the earthquake agrees with the focal mechanisms. We thereby obtain an estimation of the total stresses in the forearc and corresponding megathrust shear stresses before and after the megathrust earthquakes that is consistent with the stress-drop models and the post-mainshock fault kinematics in the forearc. The procedure is similar for all models, except for the Concepción model crossing the hypocentre location of the Maule earthquake, for which the fault kinematics in the forearc are poorly constrained by focal mechanisms (Fig. 8c). We therefore calculate the Coulomb failure stress change for the Concepción model for the initial estimate of $\tau_{b\text{-pre}}$ based on a $\mu'_b$ value of 0.03 derived from heat-dissipation models (Gao and Wang, 2014).

### 3.3 Results for the 2011 $M_w$ 9.0 Tohoku-Oki earthquake

#### 3.3.1 Sendai cross section

Fig. 9 shows the preferred model of forearc stress change due to the Tohoku-Oki earthquake for the Sendai cross section. The forearc experiences almost everywhere deviatoric compression before the earthquake (Fig. 9a). After the earthquake, the stress state is more heterogeneous and the forearc experiences deviatoric tension between 0 and ~220 km and at ~280-300 km from the trench. The extent of deviatoric tension in the model is compatible with the normal-faulting focal mechanisms between ~110-160 km from the trench (Figs. 7c and 9a), but there is one thrust-faulting focal mechanism at ~140 km from the trench. This event has a potential failure plane parallel to the plate interface and we interpret it as an event on the megathrust (cf. Nakamura et al., 2016). Alternatively, the event may indicate a local stress heterogeneity that we cannot reproduce in our models.

The modelled megathrust shear stresses before and after the Tohoku-Oki earthquake and the modelled megathrust stress drop are shown in Fig. 9b. Before the earthquake, the megathrust shear stress tends to increase with distance from the trench (i.e., depth along the fault), except near the trench, where the shear stress is comparatively high. The shear stress values relate to $\mu'_{b\text{-pre}}$ values of 0.015 to 0.022, except for the shallowest portion of the megathrust within 10 km from the trench, for which $\mu'_b$ value = 0.2. The elevated $\mu'_{b\text{-pre}}$ value near the trench is required to allow for a large stress drop on the shallowest part of the megathrust related to the large fault slip of ≥60m near the trench (Fig. 7a). Note that high stress drop near the trench may be an artifact inherited from the rupture model of Iinuma et al. (2012), a point we will revisit in section 4.2. After the earthquake, the megathrust shear stress is more heterogeneous and reaches zero at ~0-40 km and ~100 km from the trench, indicating locally complete stress drops (e.g., Brodsky et al., 2020; Hasegawa et al., 2011). The corresponding $\mu'_{b\text{-post}}$ vary between 0 and 0.025 along the megathrust. The $\Delta\mu'_b$ values vary between -0.005 and 0.022, except for the shallowest portion of the megathrust, for which $\Delta\mu'_b$ = 0.2.





**Figure 9: Preferred model of forearc stress change along the Sendai transect due to the Tohoku-Oki earthquake. See Fig. 7 for the location of the transect. (a) Differential stress and plunge of the maximum principal stress $\sigma_1$ (red: deviatoric compression, blue: deviatoric tension) before and after the Tohoku-Oki earthquake. Beach balls indicate focal mechanism solutions of aftershocks (cf. Fig. 7). (b) Modelled megathrust shear stress before (black) and after (orange) the earthquake, and megathrust stress drop (blue). (c) Coseismic change in differential stress in the forearc. (d) Coseismic change in Coulomb failure stress $\Delta CFS(\omega_{opt})$. (e) The light grey area outlines the area of positive $\Delta CFS(\omega_{opt})$. The dark grey circles show earthquake hypocentral locations.**

The stress drop on the megathrust increases and decreases the differential stress in the forearc up to 40 MPa (Fig. 9c). The differential stress increases most between 100 and 140 km from the trench. This area was close to a neutral stress state before the earthquake, such that the stress drop and following switch from deviatoric compression to deviatoric tension cause a net increase in differential stress. The largest decrease in differentials stress occurs near the trench, where the compression was comparatively high before the earthquake.



The stress change in the forearc causes large changes in Coulomb failure stress of -20 to +40 MPa (Fig. 9d). Areas of positive $\Delta CFS(\omega_{opt})$ include most of the submarine forearc within 200 km from trench and parts of the continental crust inland Japan. Areas of negative $\Delta CFS(\omega_{opt})$ include the mantel wedge at >200 km from the trench and the upper crust between ~200 and 280 km from the trench. The areas of Coulomb failure stress increase contain ~97 % of the forearc seismicity along the Sendai cross section (Fig. 9e).

### 3.3.2 Iwaki cross section

The stress state in the forearc along the Iwaki cross section is heterogenous both before and after the earthquake (Fig. 10a). Deviatoric tension occurs in most of the submarine forearc and in some parts of the continental crust inland Japan. The main effect of the megathrust stress drop is to slightly increase the deviatoric tension at >160 km from the trench and to reverse the stress state from deviatoric tension to deviatoric compression within ~60 km from the trench.

The extent of deviatoric tension in the model is compatible with the normal-faulting focal mechanisms between ~100 and ~230 km from the trench (Figs. 7c and 10a). Thrust faulting events at ~80 and ~110 km from the trench have potential failure planes parallel to the plate interface and are interpreted as megathrust events. In contrast, the thrust faulting events at ~160 and ~215 km from the trench have potential failure planes oblique to the plate interface and are likely upper-plate events. In particular, the thrust faulting at ~215 km from the trench, directly beneath the normal faulting events in the upper crust has been constrained by detailed moment tensor inversion (Yoshida et al., 2015). Both, the thrust faulting near the plate interface, at 160 km from the trench and in the lower crust, at ~215 km from the trench are compatible with the model indicating deviatoric compression in both areas (Fig. 10a).

The modelled megathrust shear stress increases with depth along the fault before the megathrust earthquake. The corresponding $\mu'_{b\text{-pre}}$ values vary between 0.02 and 0.023. There is no local peak in megathrust shear stress near the trench as for the Sendai cross section. After the megathrust earthquake, the shear stress is more heterogenous due to the stress drop, which is nowhere complete and smaller than for the Sendai cross section (Fig. 7b, d). The $\mu'_{b\text{-post}}$ and $\Delta\mu'_b$ values vary between 0.008–0.036 and -0.015–0.013, respectively. The stress drop on the megathrust induces changes in differential stress in the forearc of -12 to +16 MPa (Fig. 10a). The largest decreases and increases in differential stress occur close to the plate interface between ~50 and 120 km from the trench. Elsewhere, the differential stress changes ≤4 MPa.

The stress change in the forearc causes changes in Coulomb failure stress of -10 and +12 MPa (Fig. 10d). Areas of positive $\Delta CFS(\omega_{opt})$ include most of the forearc crust at >100 km from the trench and much of the mantle wedge between 100 and 240 km from the trench. The Coulomb failure stress mainly decreases in the submarine forearc at 40–90 km from the trench and in the mantle wedge at >240 km. The areas of Coulomb failure stress increase contain ~98 % of the forearc seismicity along the Iwaki cross section (Fig. 10e).





**Figure 10: Preferred model of forearc stress change along the Iwaki cross section due to the Tohoku-Oki earthquake. See Fig. 7 for the location of the cross section. (a) Differential stress and plunge of the maximum principal stress σ₁ (red: deviatoric compression, blue: deviatoric tension) before and after the Tohoku-Oki earthquake. Beach balls indicate focal mechanism solutions of aftershocks (cf. Fig. 7). (b) Modelled megathrust shear stress before (black) and after (orange) the earthquake, and megathrust stress drop (blue). (c) Coseismic change in differential stress in the forearc. (d) Coseismic change in Coulomb failure stress ΔCFS(ω_opt). (e) The light grey area outlines the area of positive ΔCFS(ω_opt). The dark grey circles show earthquake hypocentral locations.**

## 3.4. Results for the 2010 $M_w$ 8.8 Maule earthquake

### 3.4.1 Pichilemu cross section

The forearc along the Pichilemu transect experiences mainly deviatoric compression before the Maule earthquake and deviatoric tension within ~140 km from the trench after it. The extent of deviatoric tension in the model is compatible with the normal-faulting focal mechanisms between ~100 and 130 km from the trench (Fig. 8c and 11a). The modelled





megathrust shear stress before the earthquake increases along the fault to about ~20 MPa and then fluctuates by a few MPa before decreasing toward zero at >120 km from the trench, i.e. at a depth >40 km (Fig. 11b). The corresponding $\mu'_{b\text{-pre}}$ values vary between 0.043 and 0.02 at 5–40 km depth, and between 0.02 and 0.005 at 40–60 km depth (i.e., the coefficient of

megathrust friction tends to decrease with depth). The stress drop on the megathrust is near complete at ~60–80 km from the trench where the shear stress on the megathrust approaches zero after the earthquake. The $\mu'_{b\text{-post}}$ and $\Delta\mu'_b$ values vary between 0.006–0.04 and -0.002–0.036, respectively. It should be noted that the stress drop within 40 km from the trench is not well constrained (Fig. 8b) due to the lack of seafloor geodetic observations and is here assumed to decrease to zero toward the trench (dashed part in Fig. 11b).

The stress drop on the megathrust causes changes in differential stress in the forearc of -36 to +16 MPa (Fig. 11c). Increases in differential stress occur mainly at about 80 km from the trench where the stress drop on the megathrust is near complete and in the mantle wedge above the downdip limit of the megathrust where the stress drop on the fault is negative (Fig. 11b, c). Elsewhere, the differential stress decreases. The stress change in the forearc causes changes in Coulomb failure stress of -10 and +18 MPa (Fig. 11d). Areas of positive $\Delta CFS(\omega_{opt})$ include most of the forearc at ~60-140 km from the trench and the

mantle wedge and lower crust further inland. The areas of Coulomb failure stress increase contain ~85 % of the forearc seismicity along the Pichilemu transect (Fig. 11e).

### 3.4.2 Concepción cross section

The forearc along the Concepción cross section experiences mainly deviatoric compression before the Maule earthquake,

except between ~20 and 50 km from the trench (Fig. 12a). After the earthquake, deviatoric tension occurs up to ~70 km from the trench. Further landward the forearc remains under deviatoric compression, which agrees with two thrust-faulting focal mechanisms at about 120–130 km from trench (Figs. 8c and 12a). Both events have potential failure planes oblique to the plate interface and are interpreted as events in the mantle wedge above the megathrust.

The shear stress on the megathrust before the earthquake increases with distance from the trench to about 50 MPa at the

downdip limit of the megathrust (~170 km from the trench) (Fig. 12b). The steady increase in megathrust shear stress reflects the constant $\mu'_{b\text{-pre}}$ value of 0.03. The largest stress drop of ~8 MPa occurs at about 80 km from the trench, close to the hypocenter of the Maule earthquake (Figs. 8a and 12b). The stress drop is smaller than for the Pichilemu cross section and is nowhere complete. The $\mu'_{b\text{-post}}$ and $\Delta\mu'_b$ values vary between 0.016–0.035 and -0.005–0.014, respectively.

The stress drop mainly decreases the differential stress in the forearc by up to 16 MPa, except for the area near the trench

where the differential stress increases by up to 4 MPa. The stress changes cause changes in Coulomb failure stress from -6 to +6 MPa. The areas of Coulomb failure stress increase contain ~61 % of the forearc seismicity along the Concepción cross section (Fig. 12e).

 



**Figure 11: Preferred model of forearc stress change along the Pichilemu cross section due to the Maule earthquake. See Fig. 8 for the location of the cross section. (a) Differential stress and plunge of the maximum principal stress $\sigma_1$ (red: deviatoric compression, blue: deviatoric tension) before and after the Maule earthquake. Beach balls indicate focal mechanism solutions of aftershocks (cf. Fig. 8). (b) Modelled megathrust shear stress before (black) and after (orange) the earthquake, and megathrust stress drop (blue). (c) Coseismic change in differential stress in the forearc. (d) Coseismic change in Coulomb failure stress $\Delta CFS(\omega_{opt})$. (e) The light grey area outlines the area of positive $\Delta CFS(\omega_{opt})$. The dark grey circles show earthquake hypocentral locations.**







**Figure 12: Preferred model of forearc stress change along the Concepción cross section due to the Maule earthquake. See Fig. 8 for the location of the cross sections. (a) Differential stress and plunge of the maximum principal stress σ₁ (red: deviatoric compression, blue: deviatoric tension) before and after the Maule earthquake. Beach balls indicate focal mechanism solutions of aftershocks (cf. Fig. 8). (b) Modelled megathrust shear stress before (black) and after (orange) the earthquake, and megathrust stress drop (blue). (c) Coseismic change in differential stress in the forearc. (d) Coseismic change in Coulomb failure stress ΔCFS(ω_{opt}). (e) The light grey area outlines the area of positive ΔCFS(ω_{opt}). The dark grey circles show earthquake hypocentral locations.**





## 4 Discussion

### 4.1 Main factors promoting failure after large megathrust earthquakes

We use analytical stress solutions of the dynamic Coulomb wedge theory (Wang and Hu, 2006) and numerical finite-element
models of force balance (Dielforder et al., 2023; Wang et al., 2019) to investigate the coseismic Coulomb failure stress
changes induced by a megathrust earthquake in the forearc. Both modelling approaches allow to assess Coulomb failure
stress changes in consideration of the two main factors controlling the stress state in a forearc, i.e., the shear stress on the
megathrust causing margin-normal compression and forearc topography causing margin-normal tension. The generic models
presented in section 2 illustrate that the superposition of topographic and tectonic stresses has two major implications: First,
a forearc can experience both deviatoric tension and deviatoric compression, depending on the megathrust shear stress (Fig.
2a) (cf. Lamb, 2006; Wang and He, 1999; Wang et al., 2019). Second, the differential stress is a convex function of the
megathrust shear stress with a minimum at the neutral stress state (Fig. 2b) (cf. Dielforder et al., 2023).

The possible stress states of a forearc are crucial for understanding the conditions under which the stress drop in megathrust
earthquakes may trigger failure in the forearc. If forearcs experienced only margin-normal compression but no margin-
normal tension, the net stress drop on the megathrust would simply cause a decrease the differential stress in the forearc and
inhibit failure. Such conditions would exist in the absence of topographic stresses, i.e., if forearcs were flat (solutions for $\alpha =$
$0°$ in Figs. 2c, d and 6a). In this case, failure in the forearc could only be promoted by a negative stress drop, i.e., an increase
in megathrust shear stress.

Similarly, the stress drop on the megathrust reduces the differential stress and inhibits failure in the forearc, if the margin-
normal compression of the forearc is much larger than the margin-normal tension, i.e., if the megathrust shear stress is
sufficiently high (e.g., solutions for $\mu'_{b\text{-pre}} >0.05$ in Fig. 5a). Thus, the net stress drop in a megathrust earthquake can only
promote widespread failure in the forearc, if the megathrust shear stress before the earthquake is so low that the forearc is
either close to a neutral stress state in which the margin normal compression and margin-normal tension are similarly large
(Fig. 2a), or under deviatoric tension as, for example, along the Iwaki cross section, Japan (Fig. 10a). At this condition, the
megathrust stress drop results in an increase the deviatoric tension in the forearc, which promotes failure.

The low-stress conditions in the forearc required for a positive Coulomb failure stress change are compatible with previous
estimates of forearc stresses. Force balance analyses of global subduction zones indicates that near-neutral stress conditions
are given for effective coefficients of megathrust friction of ~0.03±0.02 (Dielforder et al., 2020; Lamb, 2006; Matthies et al.,
2024; Seno, 2009). The $\mu'_b$ values from force-balance models are consistent with estimates of $\mu'_b$ derived from other
methods, including heat-dissipation models (e.g., Bird, 1978; Gao and Wang, 2014; van den Beukel and Wortel, 1987, 1988,
Wada and Wang, 2009), constraints on pore fluid overpressures and effective stresses based on the analysis of seismic p-
Wave to s-wave velocity ratios (e.g., Moreno et al., 2014; Tsuji et al., 2014), field-observations from exhumed megathrusts
faults (e.g., Angiboust et al., 2015; Cerchiari et al., 2020; Oncken et al., 2022), and analysis of the energy budget of





megathrust earthquakes (e.g., Lambert et al., 2021). We therefore expect that near-neutral stress conditions as inferred for
Japan and Chile are common along subduction zones worldwide, which implies that most forearcs are prone to failure.

Another factor that controls failure in the forearc is the strength of faults. Coulomb failure stress models usually only describe whether a stress change promotes or inhibits failure, but do not determine the conditions that eventually allow failure, such as the critical pore fluid pressure. Likewise, our models do not describe the conditions for failure, but our findings imply that faults in the forearc must be weak enough to fail at the low differential stress magnitudes that are given in

forearcs under deviatoric tension (on average <100 MPa). Failure at such low differential stresses requires that active faults in the forearc are almost as weak as the megathrust (e.g., Dielforder et al., 2020; 2023; Wang et al., 2019; Wang and Hu, 2006; Yang et al., 2013). The low strength may be explained by high pore fluid overpressures reducing the effective stresses in the forearc or a low intrinsic strength of the fault zone. The latter may be caused by the presence of sheet silicates and the development of shear fabrics in the fault zone, which can reduce the coefficient of friction to values as low as ~0.2 (Ikari and

Kopf, 2017; Moore & Lockner, 2004; Tesei et al., 2012). The requirement of weak faults for failure also implies that their absence may inhibit failure and cause tectonic quiescence in forearc areas that experience an increase in Coulomb failure stress.

## 4.2 Robustness of the modelled Coulomb failure stress changes

The results of the finite-element models indicate that the majority of the aftershock seismicity along the studied forearc transects occurred in areas of Coulomb failure stress increase (~97-98 % for Japan, ~85 % and ~61 % for Pichilemu and Concepción, respectively). The proportions of positively stressed aftershocks (i.e., aftershocks in areas of positive $\Delta CFS(\omega_{opt})$) may be subject to uncertainties in the model parameters. For example, Ishibe et al. (2017) calculated Coulomb failure stress changes for the Tohoku-Oki, Maule, and Sumatra-Andaman earthquakes, and showed that the choice of slip

model and coefficients of friction affects the proportion of positively stressed aftershocks in their models by up to 30 %.

Uncertainties in stress drop may be large, if the earthquake-slip model used for the stress-drop calculation is constrained by onshore geodetic observations only as for the Maule earthquake (Luttrell et al., 2011; Stressler and Barnhart, 2017). There is also a large number of competing slip models for the Tohoku-Oki and Maule earthquakes, as well as models that average different slip model (e.g., Benavente and Cummins, 2013; Delouis et al., 2010; Hooper et al., 2013; Kubota et al., 2022;

Minson et al., 2014; Moreno et al., 2012; Sun et al., 2017; Wang et al., 2019; 2020; Wei et al., 2012; Yue et al., 2014). We therefore conducted supplementary finite-element models for the Sendai and Pichilemu cross sections using different stress-drop models (Fig. 13). For Sendai, we used the stress drop-model of Kubota et al. (2022) for the slip model of the same authors. The slip model of Kubota et al. (2022) includes a lower slip (~53 m) near the trench than the slip model of Iinuma et al. (2012) (~80 m), and is similar to the model of Sun et al. (2017) that quantifies the slip near the trench from high-



resolution bathymetry differences before and after the Tohoku-Oki earthquake. For the Pichilemu cross section, we used the stress-drop model of Wang et al. (2020) for the average slip model of the same authors, averaging 12 published slip models. The stress-drop models of Kubota et al. (2022) and Wang et al. (2020) yield smaller stress drops than the models of Brown et al. (2015) and Luttrell et al. (2011) (Fig. 13e, f). The absolute Coulomb failure stress changes are therefore up to 10 MPa smaller than in our preferred models (Figs. 9d, 11d). However, the Coulomb failure stresses increase and decrease in similar

forearc areas as in our preferred models and the proportions of positively stressed aftershocks increase slightly (~99 % for Sendai and ~90 % for Pichilemu) (Fig. 13c, d).

The effect of the stress drop on the spatial distribution of positive and negative Coulomb failure stress changes is comparatively small, mainly because our modelling approach requires that the post-earthquake stress state is consistent with the fault kinematics of upper-plate aftershocks. The post-earthquake stress state is therefore the same as in the preferred

models, while the pre-earthquake stress state differs. In detail, the lower stress-drops in the models of Kubota et al. (2022) and Wang et al. (2020) cause that the pre-earthquake stress states are less compressive so that larger forearc areas experience deviatoric tension, which promotes the broad increase in Coulomb failure stress. However, the pre-earthquake models may underestimate the compression of the forearc, at least for Sendai where there is evidence for thrust faulting in the forearc before the Tohoku-Oki earthquake (e.g., Nakamura et al., 2016; Wang et al., 2019).

For comparison, if the post-earthquake stress state is not constrained to be consistent with the upper-plate fault-kinematics, then the proportions of positively stressed aftershocks decrease significantly. Fig. 13g, h illustrates the Coulomb failure stress changes for the Sendai and Pichilemu cross-sections and for $\mu'_{b\text{-pre}}$ and $\mu'_{b\text{-post}}$ values that are 0.01 higher than in the models presented in Fig. 13a, b. The higher $\mu'_b$ values increase the megathrust shear stress (Fig. 13k, l) and cause that the post-earthquake stress states to become partially incompatible with the fault kinematics, so that normal faulting occurs in

areas of deviatoric compression. In consequence, the proportions of positively stressed aftershocks decrease to 56 % for the Sendai cross section and to 41 % for the Pichilemu cross section (Fig. 13i, j). An increasing discrepancy between the modelled post-earthquake stress state and the upper-plate fault kinematics further reduces the proportion of positively stressed aftershocks.

We also evaluated the effect of the coefficient of friction on the Coulomb failure stress changes by solving the

supplementary models for $\mu = 0.2$. The lower friction coefficient slightly reduces the proportions of positively stressed aftershocks (~95 % for Sendai, and 83 % for Pichilemu), i.e., the areas of positive Coulomb failure stress change become slightly smaller (see Fig. S2 in the Supplement). This finding is consistent with the analytical stress solutions (Fig. 6d), which also show that a smaller coefficient of friction decreases the Coulomb failure stress change, while the overall dependence on the stress state does not change.

Taken together, we find that the model results vary with the stress-drop model and coefficient of friction, but are most sensitive to the forearc stress state. Our finding implies that even if the megathrust stress drop and the coefficients of friction are precisely known, robust estimates of the Coulomb failure stress change may only be obtained if the megathrust shear stress and hence the total stresses in the forearc are well constrained. The high percentage of positively stressed aftershocks



in our models for Japan and Pichilemu, Chile, mainly results from our assessment of the megathrust shear stresses that we
consider well constrained within reasonable uncertainties. For comparison, the megathrust shear stresses along the
Concepción transect are less well constrained and the model results are less robust.







**Figure 13 (previous page): Supplementary Coulomb failure stress models for the Sendai cross section, Japan, (left) and the Pichilemu cross section, Chile (right). (a) Coulomb failure stress change ΔCFS($\omega_{opt}$) obtained for the stress-drop model of Kubota et al. (2022) (K22). (b) ΔCFS($\omega_{opt}$) obtained for the stress-drop model of Wang et al. (2020) (W20). (c, d) The light grey areas outline areas of positive ΔCFS($\omega_{opt}$). The dark grey circles show earthquake hypocentral locations. (d, e) Modelled megathrust shear stress before (black) and after (orange) the earthquake, and megathrust stress drop (blue). Note that the post-earthquake shear stresses in the supplementary models are identical to the one in the preferred models shown in Figs. 9b and 11b. (g–l) The same as in (a–f) but for models with higher megathrust shear stresses before and after the earthquake. The higher megathrust shear stresses are obtained by increasing the effective coefficient of megathrust friction $\mu'_b$ in the models by 0.01. See section 4.2 for discussion.**

## 4.3 Magnitude of Coulomb failure stress changes

We obtain Coulomb failure stress changes that can reach 10s of MPa, which is similar to the magnitude of the modelled differential stress and changes in differential stress (Figs. 3, 5, 9-12), and distinctly higher than previous estimates of Coulomb failure stress changes in megathrust earthquakes that are on the order of 0.01–1 MPa (e.g., Farías et al., 2011; Ishibe et al., 2017; Jara-Munoz et al., 2022; Nakamura et al., 2016; Ryder et al., 2012; Stressler and Barnhart, 2017; Terakawa et al., 2013; Toda et al., 2011b; Qiu and Chan, 2019). The higher stress magnitudes agree with previous notions that forearc stress changes induced by great megathrust earthquakes are of the same order of magnitude of the differential stress in the forearc (e.g., Chiba et al., 2013; Hardebeck, 2012; Hasegawa et al., 2011, 2012; Wang and Hu, 2006).

The high Coulomb failure stress changes result from the comparatively large stress changes that occur if the stress state switches from deviatoric compression to deviatoric tension, especially for a strong megathrust stress drop in the proximity of steep margin topography (Fig. 6a, c). Accordingly, we find that the largest Coulomb failure stress changes occur along the Sendai transect, Japan, and the Pichilemu transect, Chile, close to the shelf break at about 100 km distance from the trench (Figs. 9 and 11). In comparison, the Coulomb failure stress changes are lower along the Iwaki transect, Japan, and the Concepción transect, Chile, were the megathrust stress drop and the stress changes in the forearc are smaller (Fig. 10 and 12). Thus, the large Coulomb failure stress changes are a direct consequence of the interaction of topographic and tectonic stresses, which was not captured in previous Coulomb failure stress models.

## 4.4 Significance of the modelled stress changes as triggers of aftershock seismicity

Our modelling approach assumes that the stress state after the earthquake is compatible with the postseismic fault kinematics in the forearc. On that premise, the megathrust stress drop results in a broad increase in Coulomb failure stress, so that the majority of all aftershocks occurs in positively stressed areas (Figs. 9-13). This outcome indicates that most of the seismicity was promoted and likely triggered by the stress changes resulting from the stress drop on the megathrust. Our analysis also corroborates that much of the upper-plate aftershock seismicity of the Tohoku-Oki and Maule earthquakes occurred on faults well oriented for failure. In contrast, previous Coulomb failure stress models suggested that less than half of the aftershock



seismicity of Tohoku-Oki and Maule earthquakes can be explained by failure on optimally oriented faults (Miao & Zhu, 2012), but these models did not constrain the total stresses and stress states in the forearc.

We further find that the aftershock seismicity and the modelled Coulomb failure stress changes show congruities that

corroborate a triggering of the aftershock seismicity by the modelled stress changes, especially for the Sendai and Pichilemu transects. For Sendai, the landward extent of seismicity in the lower crust and mantle at about 160–200 km from the trench follows isolines of $\Delta CFS(\omega_{opt})$ (Fig. 9d, e). For Pichilemu, isolines of $\Delta CFS(\omega_{opt})$ at 80–130 km form landward-verging lobes that encompass the main cluster of seismicity that also verges landward (Figure 11d, e).

The consistency in the aftershock-$\Delta CFS(\omega_{opt})$ patterns does, however, not imply that there is a general, simple dependence of

the seismicity distribution on the earthquake stress changes. For example, the largest earthquake slip and the largest stress changes in Japan occurred near the Tohoku-Oki hypocentre along the Sendai cross section, but most of the aftershocks occurred ~140 km to the southwest in the coastal area near Iwaki, where the earthquake slip and stress changes were comparatively low (Figs. 7, 9 and 10). For comparison, in Chile, the largest earthquake slip and stress changes occurred near Pichilemu, ~200 km north of the mainshock hypocentre, where also most of the aftershock seismicity occurred (Fig. 8, 11,

and 12).

The differences in the aftershock occurrence likely reflect spatial heterogeneities in fault strength and proximity to failure. For Japan, our model results indicate that tectonically active forearc areas along the Iwaki cross section experienced deviatoric tension already before the Tohoku-Oki earthquake (Fig. 10a), which is consistent with upper-plate normal faulting off the coast of Iwaki in the years before the Tohoku-Oki earthquake (Hasegawa et al., 2012). Thus, normal faults along the

Iwaki transect were closer to failure than normal faults along the Sendai transect that experienced deviatoric compression before the main shock (Fig. 9a) (Wang et al., 2019). The comparatively small stress changes near Iwaki may thus have been enough to trigger normal faulting.

For comparison, the outer forearc along Sendai transect at <140 km from the trench shows comparatively little seismicity despite large increases in Coulomb failure stress (Fig. 9). The outer forearc experienced strong coseismic dilation due to the

large slip near the trench causing seaward surface displacements ≥20m (Kido et al., 2011; Sato et al., 2011), which may have caused dilatant hardening (e.g., Brace, 1978), i.e., a drop in pore fluid pressure and respective increase in effective fault strength.

The aftershock seismicity of the Tohoku-Oki and Maule earthquakes may have been further affected by processes not captured in our models. For example, aftershocks can be triggered by dynamic stress changes resulting from the passage of

seismic waves emitted by the main shock (e.g., Gomberg et al., 2004; Kato et al., 2013; Miyazawa, 2011). Coulomb failure stress changes caused by seismic waves can reach a few MPa near the earthquake hypocentre (e.g., Kilb et al., 2000; Miyazawa, 2011), which is comparable to the Coulomb failure stress change caused by the stress drop on the megathrust. Dynamic triggering in the nearfield of the earthquake should be quasi-instantaneous with the main shock (e.g., Belardinelli et al., 2003; Harris, 1998) and may have affected the immediate seismic response to the Tohoku-Oki and Maule earthquakes.

The longer-term aftershock seismicity may have been influenced by poroelastic effects and pore pressure changes (e.g.,





Cocco and Rice, 2002; Hainzl, 2004; Peikert et al., 2024; Peña et al., 2022; Terakawa et al., 2013; Yoshida et al., 2017), viscoelastic stress relaxation in the mantle wedge and lower crust (Bagge and Hampel, 2017; Becker et al., 2018; Diao et al., 2014; Sun et al., 2014), and stress changes induced by larger aftershocks ($M_w \geq 5$), as they occurred for example near Iwaki and Pichilemu (e.g., Fukushima et al., 2013; 2018; Lange et al., 2012; Ryder et al., 2012; Wimpenny et al., 2023).


## 5 Conclusions

Our analysis demonstrates that the stress change in the forearc resulting from the stress drop in a megathrust earthquake depends to first order on the stress state in the forearc. The dependence on the stress state arises because the total stresses in a forearc mainly result from the superposition of topographic stresses causing margin-normal tension and tectonic stresses

causing margin-normal compression. During megathrust earthquakes, the superposition of topographic and tectonic stresses changes, which determines the stress change in the forearc.

Topographic stresses allow the stress state in the forearc to switch from deviatoric compression to deviatoric tension, if the shear stress on the megathrust is sufficiently low. The switch in stress state is the main factor that promotes broad increases in Coulomb failure stress and widespread aftershock seismicity at the scale of the forearc. Without the stress reversal,

megathrust earthquakes have the tendency to stabilize the forearc and inhibit aftershock seismicity. A switch in stress state is supported if the megathrust is very weak and the forearc is close to a neutral stress state (margin normal compression ≈ margin normal tension) before the earthquake. Near-neutral stress conditions have been inferred for most global subduction zones (Dielforder et al., 2020; Gao and Wang, 2014; Heuret et al., 2011; Lamb, 2006). Thus, the mechanisms evaluated here are crucial for assessing the geohazard at convergent margins.

The dependency of the stress change on the stress state introduces the challenge that the total stresses in the forearc must be constrained in order to determine the Coulomb failure stress change induced by a megathrust earthquake. We show that the total stresses before and after the earthquake may be constrained by moment tensor solutions of upper-plate aftershocks and estimates of the megathrust stress drop in the mainshock. Thus, the availability of earthquake moment tensor solutions and stress drop estimates is crucial for evaluating seismic hazards at active margins. This underlines the importance of installing

high-quality geophysical networks, such as those in Japan. If detailed geophysical observations are not available, Coulomb failure stress changes may still be estimated, for example by using for the stress calculations an effective coefficient of friction for the megathrust determined by other means. However, this approach can introduce uncertainties and lead to less accurate models, as likely reflected in our model for the Concepción cross section in Chile.

The benefit of constraining the total stresses before and after the earthquake is that it allows a better estimate of Coulomb

failure stress changes and provides insights into the stress conditions that promote aftershock seismicity. Our models for Japan reveal differences in the preseismic stress conditions along the Sendai and Iwaki cross sections, which may explain why the comparatively small stress changes near Iwaki could trigger intense aftershock seismicity. In contrast, the intense



aftershock seismicity near Pichilemu (Chile) was dependent on the large stress changes that resulted from large slip and stress drop on the megathrust ~200 km north of the mainshock hypocentre.

Finally, our models illustrate the importance of forearc mechanics for understanding Coulomb failure stress changes and aftershock triggering, but they are currently limited in their application to two-dimensional cross sections normal to the plate margin. Future work will therefore include the development of three-dimensional models, which will account for differences in continental-margin relief and total stresses along strike of the margin.

**Appendix A**

The stress solutions for an elastic-perfectly Coulomb plastic wedge can be written in terms of effective stresses as (Wang and Hu, 2006)

$$\bar{\sigma}_x = m(1-\lambda)\rho g z \cos\alpha \qquad (A1a)$$

$$\bar{\sigma}_z = (1-\lambda)\rho g z \cos\alpha \qquad (A1b)$$

where

$$m = 1 + \frac{2[\tan\alpha' + \mu'_b/(1-\lambda)]}{\sin 2\theta[1 - \mu'_b/(1-\lambda)\tan\theta]} - \frac{2\tan\alpha'}{\tan\theta} \qquad (A2)$$

and

$$\tan\alpha' = \frac{1-\rho'}{1-\lambda}\tan\alpha \qquad (A3)$$

with $\theta = \alpha + \beta$, and $\rho' = \rho_w/\rho$. The total stresses $\sigma_x$ and $\sigma_z$ can be obtained by solving equation (A1) for $\lambda = 0$.

The principal stresses can be expressed as (Dahlen, 1990)

$$\sigma_1 = \sigma_z - 0.5(\sigma_z - \sigma_x)(1 + \sec\psi_0) \qquad (A4a)$$

$$\sigma_3 = \sigma_z - 0.5(\sigma_z - \sigma_x)(1 - \sec\psi_0) \qquad (A4b)$$

where $\sigma_1$ and $\sigma_3$ are the greatest and least compressive stresses, respectively, and $\psi_0$ is the angle between the surface of the wedge and the axis of $\sigma_1$. Angle $\psi_0$ is determined from the following equation (Wang and Hu, 2006)

$$\frac{\tan 2\psi_0}{\cos\varphi^p \sec 2\psi_0 - 1} = \frac{\tan\alpha'}{1+\eta} \qquad (A5)$$

where

$$\varphi^p = \arcsin\sqrt{\frac{(m-1)^2 + 4\tan^2\alpha'}{(2\eta+m+1)^2}} \qquad (A6)$$

The cohesion gradient $\eta$ is a dimensionless constant that allows to account for wedge cohesion (Zhao et al., 1986). Equation (A5) may be rewritten in explicit forms as

$$\psi_0 = \frac{1}{2}\left(\arcsin\left(\frac{\sin\alpha''}{\sin\varphi^p}\right) - \alpha''\right), \psi_0 \leq \pi/4 - \alpha''/2 \qquad (A7a)$$

$$\psi_0 = \frac{\pi}{2} + \frac{1}{2}\left(\arcsin\left(\frac{\sin\alpha''}{-\sin\varphi^p}\right) - \alpha''\right), \psi_0 > \pi/4 - \alpha''/2 \qquad (A7b)$$



where

$$\alpha'' = \arctan\left(\frac{\tan \alpha'}{1+\eta}\right) \tag{A8}$$

The wedge enters a critical state if

$$m = m^c = 1 + \frac{2(1+\eta)}{\csc \varphi \sec 2\psi_0^c - 1} \tag{A9}$$

where $\varphi = \arctan \mu$ and $\psi_0^c$ is the angle of between the axis of $\sigma_1$ and the surface slope at critical state (Wang and Hu, 2006, Zhao et al., 1986). Angle $\psi_0^c$ is determined from the following equation similar to Equation (A5)

$$\frac{\tan 2\psi_0^c}{\cos \varphi \sec 2\psi_0^c - 1} = \frac{\tan \alpha'}{1+\eta} \tag{A10}$$

The effective coefficient of megathrust friction at neutral stress state can be calculated as (Wang and Hu, 2006)

$$\mu'_{b-N} = \frac{(1-\lambda) \cos 2\theta}{\cot \alpha' + \sin \theta} \tag{A11}$$


*Code availability:* The finite-element models were calculated, processed, and plotted using the commercial software packages ABAQUS (Abaqus, 2014), MATLAB (The MathWorks Inc., 2022) and the Matlab tool Abaqus2Matlab by Papazafeiropoulos et al. (2017). The maps in Fig. 7 and 8 were produced with GMT (Wessel et al., 2019). Colour schemes

follow 'Scientific colour maps' (Crameri et al., 2020).

*Data availability.* The results of the finite-element models are archived at https://doi.org/10.25835/hbxp5xp0.

*Author contribution.* AD: Conceptualization, investigation, formal analysis, methodology, visualization, Writing – original

draft, preparation. GMB: formal analysis, methodology, visualization, writing – review and editing. AH: methodology, writing – review and editing.

*Competing interests.* The authors declare that they have no conflict of interest.

*Acknowledgements.* We thank Carlos Peña and Jonathan Bedford for discussion, and Kelin Wang and Karen Luttrell for providing the stress-drop data for the Tohoku-Oki and Maule earthquakes, respectively.

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
