# Peer review of "Importance of forearc topography for the triggering of aftershocks of megathrust earthquakes: Insights from mechanical models and the Tohoku-Oki and Maule earthquakes"

_EGUsphere, 2024_

## Author Comment (AC1)

The logic of the study is simple but the writing sometimes makes it convoluted and unnecessarily complicated. The worst example is the first paragraph of the Conclusions. Besides being unclear whether "stress state" refers to the state before or after the earthquake, or both, what is said here is incorrect at face value. In the linear system considered in this study, stress change is uniquely determined by fault stress drop, with no relation with the absolute state of stress before or after the event. If by "stress state" you actually mean "Coulomb stress", then nothing new is being said here, as Coulomb stress is always known to depend on pre-earthquake stress state. This paragraph should be deleted. There are similar situations in other parts of the paper. I leave it to the authors to check them out.

We have deleted the paragraph and revised other parts of the manuscript accordingly.

I think the discussion in Section 2.2 is unnecessarily detailed and thus distracting. Much of it (with figures) should be moved to the Supplement. The practical difference between the two definitions is very small, and I am not sure if the new definition has actual scientific advantage (see specific comment on line 230-231 below).

We have simplified the section 2 and reduced the number of figures. We have detailed some implications of the dynamic Coulomb wedge theory (DCWT) to better explain the advantage of our CFS calculations. Further details on the revision are provided in our response to the comment on line 230-231 below.

It is difficult to understand the finite element model results given the way they are presented. For example, in Fig. 9d, presumably the positive Coulomb stress in the lower crust in 200 – 280 km distance is for normal faulting, because Fig. 9a shows a steepening of sigma_1. But such steepening can also indicate less compressive stress instead of extensional stress. I am sure that the rigidity contrast across the model Moho is responsible for the lower-crust positive Coulomb stress, but I cannot tell how. It is not possible to understand the results based on the information shown in these plots (see specific comments on Fig. 9 below).

We have revised the presentation of the model results and now illustrate the stresses before and after the earthquake, as well as the coseismic incremental stress change, in terms of deviatoric stress and using stress crosses, similar to Figure 5 in Wang et al. (2019). The stresses in the models are always compressive; real tensional stresses do not occur. The lower crust in 200-280 km distance remains under deviatoric compression, as now shown by red stress crosses or red dashes in the previous version of the figure. It is right, that the rigidity contrast across the Moho influences the Coulomb failure stress change (DCFS). Without the rigidity contrast the DCFS is slightly negative in that region. We now address the rigidity contrast in the discussion section and provide supplementary models without the rigidity contrast (Figs. S9-S11). The new Table 1 summarizes how different model parameters affect the percentage of positively stressed aftershocks.

Other comments by line numbers:
58-- requires to account for -> requires accounting for
We have corrected the syntax.

79-- alpha is slope angle, not slope.
We have replaced "slope" with "slope angle" where necessary.

102-- I know that the author understands this but is trying to avoid bringing up more complications. Unfortunately, it cannot be avoided. Because Dahlen's lambda_b is a

small-taper approximation, so is his mu_b', and therefore his solution is not exact (as explained in Wang et al., 2006 GRL). Only if mu_b' is properly defined, will the solution be exact.
We have added this detail (lines 96-102).

138-- Because "dynamic weakening processes" is used to describe processes under high rate friction today, it is better to say "coseismic weakening" here.
We have changed "dynamic weakening" to "coseismic weakening".

168-170. It is incorrect to use the absolute value operator here. For example, if tau flips direction such that tau_post = -tau_pre, this equation would incorrect yields a delta(tau) = 0 instead of 2*tau_pre. One just has to specify that tau is in the direction favouring slip as in King et al. (1994), then -tau will resist slip.
The calculation of DCFS differs for optimal failure planes and for faults of specified orientation. King et al. (1994) describe both approaches. The CFS on a failure plane is defined for the shear stress magnitude, i.e., for the absolute value (e.g., Oppenheimer et al., 1998; Reasenberg and Simpson, 1992; King et al., 1994; Harris, 1998). When calculating DCFS for optimal failure planes, as in our study, the use of the absolute value operator is correct, because the sign of the shear stress only reflects the sense of shear, that differs for the two failure planes. The change in tau in the example of the reviewer implies that the failure planes change their sense of shear (the one from sinistral to dextral, and the other from dextral to sinistral), but it would not bring the failure planes closure to failure. Thus, delta_tau should be zero. However, if DCFS is resolved on a fault of specified orientation, then the sign of the shear stress is defined as mentioned by the reviewer.
We understand that the latter calculation of DCFS is more common in the literature as it can be carried out without considering total stresses. Many readers may be more familiar with the approach, and the difference to our approach is essential. To acknowledge this situation, we now illustrate both approaches of calculating DCFS in section 2 (see revised Fig. 4) and briefly explain why the DCFS for optimal failure planes is the more appropriate choice for our study. In this regard, we have also detailed implications of the DCWT at the beginning of section 2. It should now become clear, that our DCFS values reflect the same tendencies as changes of mu_b in the mu_b-lambda space of the DCWT.

Fig. 4. The plots in (a) and (b) are switched by mistake and therefore contradict their headings at the top.
The figure has been removed.

230-231. Not a valid argument. The conventional definition does not require the knowledge of pre-existing weakness and stress anisotropy either.
The argument has been removed.

314-- I am curious how the code prevents numerical instability at large (e.g. 250 km) depths if gravity is applied as a body force. Because of the very large lithostatic stress, differences between principle stresses are beyond computer precision.
Abaqus allows using double-precision (with 64-bit word length) for the model execution, so we cross-checked and confirmed that the computer precision used for the models is sufficient.

385-389. I suspect the large mu_b'=0.2 adds much push against the upper plate. Without it, would mu_b' for the rest of the fault be larger than 0.015 to 0.022?
The mu'_b-pre values for the rest of the fault would not be larger, because the values are obtained by adding the Delta_mu'_b values (calculated from the mean stress drop) to the mu'_b-post values. The mu'_b-post are determined by finding the values required for deviatoric tension in areas of normal faulting.

Fig. 9a. I find it difficult to understand the model results because the display contains no information on shear stress and stress magnitude. Is it possible to plot stress crosses scaled with stress magnitude?
We now illustrate the stresses by stress crosses scaled with magnitude as in Wang et al. (2019).

Fig. 9c. Differential stress without direction misses important information. How do we know whether the change promotes compressive or extensional failure?
The type of promoted faulting depends on the state of stress. Normal faulting is promoted if the wedge is under deviatoric tension (plunge of s1 >45°); thrust faulting if the wedge is under deviatoric compression (plunge of s1 <45°). The plunge of s1 was indicated in panel a) and color coded (red: deviatoric compression, blue: deviatoric tension). We have now adjusted the illustration following Fig. 5 in Wang et al. (2019). Note that we also distinguish a plunge of s1 between 40-50°

Fig. 9e. Are the observed earthquakes in 200 – 280 km distance normal-faulting events? Presumably the positive Coulomb stress in the lower crust in this region shown in Fig. 9d is for normal faulting, because Fig. 9a shows a steepening of sigma_1. But such steepening can also indicate less compressive stress instead of extensional stress. I am sure that the rigidity contrast across the model Moho is responsible for the lower-crust positive Coulomb stress, but I cannot tell how.
The Coulomb failure stress changes are for optimal failure planes. Thus, it depends on the stress state and related plunge of s1 whether normal or thrust faulting is promoted (see previous response). The area in 200-280 km distance is under deviatoric compression such that thrust faulting is promoted. Note, however, that the plunge of s1 is close to ~40-45° in some areas, in which case both normal and thrust faulting may be supported. There are no Japan Meteorological Agency focal mechanism solutions for the earthquakes in 200-280 km distance. The catalogue of Yoshida et al (2012) show some reverse faulting near the coast, consistent with the stress state in our model.

415-- If it was extensional also before 2011 as said later in the text, it should be explained here. The figure only shows events after 2011. Nakamura et al. (2016) showed a mixture of reverse and normal events before 2011 in this area.
We have added this information (lines 406-409): "The stress state in the forearc along the Iwaki cross section is heterogenous before the earthquake (Fig. 8a). Most of the submarine forearc is in a neutral stress state (plunge of $s_1$ 40-50°) or under deviatoric tension, which is compatible with mixed reverse and thrust faulting reported for the years before the Tohoku-Oki earthquake (e.g., Hasegawa et al., 2012; Nakamura et al., 2016; Yoshida et al., 2012)."

427-- The small mu_b-pre values used here may be needed to keep the stress drop low so that is does not exceed the values shown in Fig. 10b. However, there was large shallow afterslip in this area, and the total stress drop responsible for the

aftershocks is larger than the coseismic stress drop shown in Fig. 10b. Iinuma used GPS over a much shorter time window that reflects mostly coseismic change, but the aftershocks are affected by the afterslip which continues to relieve stress on the megathrust over a longer time.

The mu_b-pre are obtained by adding Delta-mu_b to the mu_b-post values as explained in section 3.2. Thus, the mu_b-pre values do not need to be low because of the stress drop. The values are low because the stress drop is small and because the mu_b-post values need to be low to cause deviatoric tension in areas of normal faulting (especially near Iwaki). We agree that stress release due to afterslip may bias the inferred extent of deviatoric tension. We now address this aspect at the end of section 3.2 (lines 335-342). "… it should be noted that most of the post-mainshock focal mechanisms indicate normal faulting, some of which may have been caused by afterslip and aftershocks on the megathrust in the postseismic period (e.g., Bedford et al., 2016; Nakamura et al., 2016; Sun et al., 2014). Such events may record stress release on the megathrust in addition to the coseismic stress drop and influence our assessment of the post-seismic stress state. We expect this potential effect to be small on our calculations because the normal faulting started soon after the mainshocks (e.g., Farías et al., 2011; Lange et al., 2012; Yoshida et al., 2012; Japan Meteorological Agency) and affected always the same forearc areas in the first postseismic year (Figs. S3-S4 in Supplement)."

511-512. The second point is not very useful. More fundamental is the plunge which shows tension vs. compression.

We agree and have simplified the entire paragraph.

514 onward. Poor writing. Reverse the sequence by first saying flat surface can only allow compression, both before and after an earthquake…

The sentence has been removed.

521-522. can promote … only if …

The sentence has been removed.

540-- There should be a distinction between pervasive and local failure. See discussion in Section 4.4 of Wang et al. (2019). In my view, the lack of recognition of potentially very large, multi-scale heterogeneity is the biggest shortcoming of Coulomb stress analysis as is commonly conducted. I do not ask the authors to solve this problem in this work, but some qualitative discussion will be useful.

We have added some qualitative arguments (lines 516-524): "The requirement of weak faults for failure also implies that their absence may cause tectonic quiescence even though the Coulomb failure stress increases. High pore fluid overpressures may be difficult to sustain through time and over large areas such that only small fractions of the forearc lithosphere may be close to failure. Stress changes caused by megathrust earthquakes may therefore preferentially drive small earthquakes (Wang et al., 2019). Consistently, the vast majority of the earthquakes investigated in this study have low magnitudes of about 2.5-3.5 and record local failure on small faults. However, the aftershock seismicity of both mainshocks also included damaging earthquakes with magnitudes of 6.6-7.0 inland Japan near Iwaki (Fig. 5a) and in the coastal region near Pichilemu, Chile (Fig. 6a). The large magnitude aftershocks occurred in earthquake clusters affecting the entire crust down to 20 km (Iwaki) and 35 km (Pichilemu) depth, which shows that megathrust earthquakes can cause pervasive failure in the interior of forearcs"

576-- cause -> would necessitate
We have changed "cause" to "would necessitate".

589-590. I have a hard time finding what mu value was used for all the other models. Was it 0.7? It should be prominently stated somewhere, and the reader should be reminded here again.
All main model results are obtained for mu = 0.7. We have added this information to the figure caption and state it more prominently in the first paragraph of section 4, which now provides basic information on the presented model results.

---

## Author Comment (AC2)

While the focus of the introduction is clearly the role of topography on the pre-stress conditions, it seems to me that it is not the case for the whole paper. The first part indeed focuses on topography, from a simple analytical perspective which I like very much, but the modeling part introduces additional complexities (rheological, geometry) which are not necessarily clearly introduced in the first place. Furthermore, the limits on the sources of pre-stress are not necessarily discussed which makes the whole description of the objectives not that clear to me. For instance, the remanence of stresses from previous earthquakes or previous cycles is not discussed while it could be of similar orders of magnitude of the other sources mentioned currently.

We have revised different parts of the manuscript to better explain the relevance and meaning of topography and pre-stress. Section 2 now addresses implications of the dynamic Coulomb wedge theory (DCWT) and we better explain that topographic relief is a prerequisite to cause stress changes by a megathrust stress drop that support failure in the upper plate (new Fig. 2). Regarding the numerical models, we have revised the discussion of the modelling results and included two paragraphs in section 5.3 that explicitly address the importance and effects of topography (lines 591-610). We have further conducted additional models to address the impact of model parameters, including the different material properties of crust and mantle and the presence/absence of topography. The detailed modelling results are included in the supplement (Figs. S5-S14). The effect of the parameters on the percentage of positively stressed aftershocks is summarized in the new Table 1 included in the main text.

We agree with the reviewer that the stress state before the earthquake will be a consequence of previous earthquakes and earthquake cycles. The calculation of the coseismic Coulomb failure stress change (DCFS) does not depend on how the pre-earthquake stress state was achieved and our models (as well as the DCWT) do not allow to assess processes in the previous interseismic period or during previous earthquake cycles. We now explain this aspect in the method section 3.2 (lines 329-333): "We thereby obtain an estimation of the total stresses in the forearc and corresponding megathrust shear stresses shortly before and after the megathrust earthquakes that is consistent with the stress-drop models and the post-mainshock fault kinematics in the forearc. The forearc stress states in the interseismic periods before and after the mainshock are not determined and do not influence the calculation of the coseismic Coulomb failure stress change."

The revised description of the DCWT further includes a simplified statement (adopted from Wang & Hu, 2006) that briefly explains how the interseismic period conditions the immediate pre-earthquake stress state (lines 125-128): "Over the interseismic period, the shear stress on the seismogenic megathrust increases progressively such that the maximum compression of the wedge occurs toward the end of the earthquake cycle. During megathrust earthquakes, the shear stress on the plate interface decreases abruptly due to coseismic weakening processes (e.g., Kanamori and Brodsky, 2004; Scholz 1998; Wang and Hu, 2006)."

Finally, we explain in section 3.1 that the pre-stress applied in the numerical models in step 1 is the lithostatic stress and is only applied to ease the computation of total stresses.

There is no comparison between the potential to explain the location of aftershocks using the classic coulomb approach (semi-elastic half space following King et al 1994) with the various refinements proposed here. While the authors state that

topography is a "first order" contribution, there no elements in the paper supporting that assertion. The first order contribution to aftershocks is not the pre-stress but the earthquake itself and it would be great to be able to quantify the effect of each of the improvements brought to the calculations. One way would be to simply turn off gravity in the models, but also see what happens without the rheological heterogeneities and quantify the performance of the different models. If the parameterization proposed by the authors is indeed of importance, it should definitely greatly improve the overlap between the positive CFS regions and the aftershock distributions. I am asking this because simple CFS seems to work to some extent for strike slip earthquakes where topography does not play a major role, compared to subduction zones.

We now better explain the importance of topography in sections 2 and 5 as described in our previous response. We have also carried out models without topographic stresses, i.e. models with a flat surface and without water loads (turning gravity off is not an option in the force-balance models because without gravity there is no stress, as all the stresses result from gravity). Without topography, the Coulomb failure stress change is negative almost everywhere in the forearc (Figs. S12-S14 in the Supplement). The reason for this and the difference of our DCFS calculation for optimal failure planes to the common approach of calculating DCFS for faults of specified orientation is explained in section 2.

We have further added to the discussion a comparison of our modelling results with the outcome of previous models (section 5.4, lines 642-673). We address that the percentages of positively stressed aftershocks are higher than in previous models (≥97% vs. 60-70% for Japan, and 64-87% vs. <50% for Chile). We also address similarities and differences between the modelling approaches.

I havent found in the paper the notion of attribution of the earthquakes to a sequence of aftershocks. I guess the authors have carefully taken care of this aspect but are all the events used here really aftershocks and how is the selection performed? Is it simply over a given time period after the mainshocks and if so, is the duration of that period of importance? I know that there is no time dependent processes in the CFS calculation but there is a time dependency in the aftershock distribution and over a long time, some "interseismic seismicity" should show up in the dataset, polluting the interpretation in this case.

We now provide arguments for considering the seismicity as aftershocks (lines 583-593, Figs. S3-S4 in Supplement). The arguments include: 1) increased seismicity rates following the mainshock, 2) the investigated seismicity appears in areas that were inactive before or shows little seismicity, 2) the seismicity rates are highest immediately after the mainshock and decay at a power-law like rate afterwards, 3) the location of seismicity remains similar over time.

Minor points:
There is two sections 2.2
We have corrected the section numbering.

Point P1 (figure 2) is not indicated on figure 1
The Coulomb wedge model and the sketch is scale independent; we therefore prefer to do not indicate the point. More importantly, we realized that it is more accurate to state that these are "stresses at 10 km depth" rather than at a coordinate X,Y

because the stress sigma_x does not depend on horizontal distance (equation A1a). We have adjusted the text and figures accordingly.

I am not convinced by the need for figure 4 since it simply explains what CFS is. It certainly is a nice textbook figure, but I don't see how it brings new elements to the discussion. In general, removing elements that are known or separating them clearly from what is new would certainly help clarifying the objectives of the paper.
We have removed the figure.

Line 515: Some words are missing in this sentence
We have simplified the entire section and the sentence has been removed.

Line 525: Some words are also missing here.
We have simplified the entire section and the sentence has been removed.

The first sentence of the conclusion is quite awkward. The stress change does not depend on the initial stress since it only depends on the stress drop. If you mean the CFS, then yes, but please clarify.
We have removed the sentence and other misleading formulations in the manuscript.